# Assessment of Biosecurity Practices and Its Status in Small- and Medium-Scale Commercial Poultry Farms in Arsi and East Showa Zones, Oromia, Ethiopia

**Dereje Tsegaye [1,2,\*], Berhan Tamir [2] and Getachew Gebru [3]**

[1] Department of Animal Science, College of Agriculture and Environmental Science, Arsi University, Oromia P.O. Box 193, Ethiopia

[2] Department of Animal Production, College of Veterinary and Agriculture, Addis Ababa University, Addis Ababa P.O. Box 1176, Ethiopia; berhantamir@yahoo.com

[3] Partnership—MARIL Ethiopia; Chair, Agriculture Working Group and Ethiopian Academy of Sciences, Addis Ababa P.O. Box 32228, Ethiopia; ggebru09@gmail.com

**\*** Correspondence: deretsegaye683@gmail.com; Tel.: +251-913183901

**Abstract:** Disease prevalence and seasonal outbreaks are challenging the poultry industry in Ethiopia. Proper and sustainable implementation of biosecurity practices is important to reverse such problems. This study was conducted in commercial poultry farms in two zones of Ethiopia to investigate farm characteristics, implementation of biosecurity practices, and biosecurity status (BS) using a structured questionnaire. The variables were grouped into three biosecurity factors, including conceptual, structural, and operational biosecurity, based on their homogeneity. Descriptive and inferential statistics were used to summarize the results. Most commercial farms were owned by males (69.7%). The majority of the farms (40.3%) were located at a distance <50 m from residential areas. Farm owners do not provide biosecurity training to their employees (68.8%), which results in poor biosecurity implementation. The mean conceptual, structural, and operational BS were $50.4 \pm 11.62$, $63.27 \pm 10.51$, and $44.69 \pm 13.04$, respectively, indicating operational biosecurity measurements were less implemented. Overall, the BS indicated that 40.7% of the farms have BS < 50% questing for interventions. Farm characteristics and biosecurity measurements were positively associated with BS, which shows substantial room for improvement. Owners' education, occupation, experience, farm flock size, and training were significantly associated with BS ($p < 0.05$). A disease prevention strategy through biosecurity improvement is an economical means for controlling poultry disease prevalence.

**Keywords:** biosecurity; diseases; poultry; farms; biosecurity status

## 1. Introduction

Livestock plays a crucial role in the livelihoods of the farming community in Ethiopia. The livestock sector has social, economic, and cultural values for Ethiopians and is highly integrated with crop agriculture. Poultry is the dominant livestock species in Ethiopia, estimated at 57 million chickens next to cattle which is 70 million [1]. Among the huge number of chickens in Ethiopia, laying hens contributes the lion shares (34.26%), followed by chicks (32.86%), pullets (11.35%), cocks (11.12%), cockerels (5.74%), and nonlaying hens (4.59%). About 78.85% of the chicken were indigenous, 12.02% hybrid, and 9.11% were exotic [1]. Regarding productivity, the average number of eggs laid per hen per laying period was about 13 eggs, 48 eggs, and 128 eggs for indigenous, hybrids, and exotic chickens, respectively. The average annual egg production was about 369 million in the year 2020 [1], and poultry meat production in the year 2016 was 13,000 tones contributing only 2%, 0.2%, and 0.01% of the total poultry meat outputs of East Africa, Africa, and the rest of the world, respectively [2]. Livestock farming in Ethiopia has social, economic, and livelihood values and plays a crucial role in food and nutritional security. Globally,

the nutrition, food security, livelihoods, and resilience of hundreds of millions of people depend on animal products. Livestock contributes one-third of the protein that people consume in the world [3]. Worldwide, 40% of all agricultural income comes from livestock. In Ethiopia, the contribution of livestock accounts for about 15–18% of the total GDP and 40–49% of the agricultural GDP, excluding the value of animals as draught power, manure, and transport of people and products [4]. The dominant poultry production system in Ethiopia is the traditional backyard system characterized by poor management, biosecurity, health, and productivity, dominated by local chickens. There are a few numbers of small-, medium-, and large-scale commercial poultry farms in Ethiopia, and most of them are located near the capital or regional towns [2]. Very recently, business-oriented small-scale poultry enterprises were emerging to enhance the benefit of the sector. The number of chickens supplied to the market for sale and slaughtering in Ethiopia was 15.8 mil and 14.3 mil in 2020, respectively, which is very small relative to the chicken population of the country, whereas the annual mortality rate for various reasons was very high, nearly 39 mil [1]. The per capita chicken egg and meat consumption in Ethiopia were 0.36 kg and 0.66 kg, respectively, which is the lowest in Africa and in East Africa average of 1.03 kg of egg and 1.64 kg of meat in the year 2013 [2].

Among the multidimensional challenges of poultry enterprises, high poultry disease prevalence in commercial farms and household level is a critical constraint in Ethiopia for reasons such as poor farm biosecurity, inadequate health facility, lack of prevention strategies, and poor access to vaccines. Biosecurity is one of the root causes of disease prevalence and outbreaks, though its compliance is usually poor in all animal production systems around the world [5]. Biosecurity is a set of practices and strategies implemented to control, bind, and prevent the entry and transmission of infectious diseases in poultry farms and facilities [6–8]. A comprehensive biosecurity program needs to consist of an ordered set of conceptual, structural, and operational elements that are designed to stop the spread of infectious diseases both inside and across farms and facilities. Biosecurity management should be implemented primarily beginning with site selection for farm facilities, secondly with farms' physical factors (layout, drainage, and fencing), and thirdly with routine procedures such as bioexclusion and spread (biocontainment) of infection within a facility. Any disease control program should regularly assess and practice such procedures and actions. In Ethiopia, most small and medium commercial farms operate under low levels of biosecurity, which increases the risk of the spread of infectious diseases. Therefore, the current study was designed to investigate biosecurity management practices and assess biosecurity status (BS) in small- and medium-scale poultry farms in the study area.

## 2. Materials and Methods

### 2.1. Study Area

The study was conducted in the southeastern part of Ethiopia, particularly in the Tiyo, Dodota, and Hetosa districts of The Arsi zone and Adama, Bishoftu, and Boset districts of East Showa zones, Oromia regional states, Ethiopia. The altitude of the study areas ranges from 1458 masl to 2490 masl, and the latitude ranges from 7.9° N lat and 39.1° E long to 8.7° N lat and 39° E long with minimum and maximum temperatures of 8.4 °C and 31.7 °C, respectively.

### 2.2. Study Population

The targeted population of the study was all small- and medium-scale commercial poultry farms established by private-, micro-, and small-scale enterprises (MSEs) and cooperatives that raise exotic breeds of chickens for egg, broiler, and pullet production. Small farms own <1000, whereas medium farms own 1000–10,000 exotic/cross-breed chickens. In total, 221 poultry farms having a minimum and maximum flock size of 50–5000 were visited.

*2.3. Study Design and Sampling*

As a research design, a cross-sectional survey was carried out from December 2021 to February 2022 in 221 small and medium commercial poultry farms. Lists of such commercial poultry farms were collected from respective Woreda Livestock and Fisheries Development offices. In this study, we have used a checklist-based one-to-one interview method to collect data. Before proceeding to data collection, we have tried to discuss with poultry farm owners and farms' operators. We disclosed the objectives of the study, the data to be collected, and how the data were collected. In addition, we told them that the study does not harm their farm and they do have the right to withdraw at any time if they are not interested in participating. Finally, we have asked them for their willingness to provide us with the necessary data and allow us to visit their farms at some points. As a result, verbal consent was secured prior to data collection at the sampling stage, which could be witnessed through personal contact with sample poultry farm owners at random. Then, the data was collected from those farm owners who were willing to provide the necessary data based on informed consent by dropping those unwilling farm owners and operators.

*2.4. Questionnaire Development*

A semi-structured questionnaire was designed mainly on biosecurity practices adopted by commercial poultry farms focusing on conceptual, structural, and operational biosecurity measures/practices and some farm characteristics. The conceptual biosecurity framework includes the location of the farm facilities, distances from residential areas, roads, and other facilities, the existence of standing water, materials used and housing type, and the like. The structural biosecurity framework comprises issues related to the existence of barriers for the entry of infectious agents such as foot baths, farm gates, fences, vehicle tire baths, wild birds, no purchase of chicken, no access of rodents to feed, absence of pet animals in the farm, etc. Operational biosecurity measure includes precautions in relation to employees such as clothes, shower, glove, masks, footwear, visitors' cloth, and other operational activities such as partial depopulation, chicken examination, sanitary practices, and so on. The questionnaire associated with farm characteristics consists of farm type, farm size, chicken type, breed, farm capacity, productivity, production cycle, and the like.

*2.5. Data Collection*

Data collection was carried out using semi-structured questionnaires developed for this purpose comprising pertinent closed and open-ended questions that help to gather all the information regarding biosecurity measures and farm characteristics. The questionnaire was pretested before the final survey in order to refine the questionnaire to make it clear, understandable, and complete. Finally, a face-to-face personal interview was carried out with farm owners in the case of a private farm, managers/employees in the case of MSE, and cooperative-type farms.

*2.6. Data Management and Analysis*

In order to score biosecurity, variables in the questionnaire received an individual score of 0 (for a total absence of preventive measures) and 1 (for full presence of preventive measures), according to [9,10]. The variables were categorized into three homogenous groups depending on the nature and similarity of the variables in their influence on the potential risk of poultry disease introduction into a given poultry farm, such as conceptual, structural (facilities and equipment), and operational biosecurity factors [6,10–12]. Then, mean BS and percentages of BS were computed for structural, conceptual, and operational biosecurity measurements. The computed biosecurity scores (BS) were compared with the standard biosecurity rating "Good" if the BS of the farm was above 50% and "Poor" if the BS of the farm was less than 50% [6]. Accordingly, the final generated data were entered into SPSS version 22 statistical software and analyzed using descriptive statistics such as frequency, mean, and percentages. Analysis of variance (ANOVA) and t-test were

computed to determine the significant differences between variables. Pearson's chi-square was computed to determine the relationships between farm characteristics and BS.

## 3. Results

### 3.1. Demography of Farm Owners

A total of 221 poultry farms with different flock sizes were visited during this survey. Most of the farms (121) were privately owned, 82 farms were owned by micro- and small-scale enterprises, and 17 poultry farms were established as cooperatives. In most cases, the owners were the managers of the farms, of which the majority (69.7%) of the farms were owned/managed by males. Thirty-two percent of the respondents were less educated (illiterate or up to grade 4), which impacted the success of a business and the entrepreneurial skill of individuals.

The majority of the owners (80.5%) have previous work experience in poultry production without formal education (Table 1). Farm owners (79.7%) have gotten technical training opportunities not specific to biosecurity by government and nongovernmental organizations, and 20.4% did not obtain any training. Among these owners, only 60.6% let external professional employees supervise their farms.

**Table 1.** Sex, age, educational level, and experiences of the respondents.

| Variables | | Arsi Zone N (%) | East_Showa N (%) | Total N (%) |
|---|---|---|---|---|
| Sex of the respondents | Male owned | 70 (70%) | 84 (69.4%) | 154 (69.7%) |
| | Female owned | 30 (30%) | 37 (30.6%) | 67 (30.3%) |
| Age of the respondent (years) | 15–30 | 26 (26%) | 26 (21.5%) | 52 (23.5%) |
| | 31–45 | 53 (53%) | 69 (57%) | 122 (55.2%) |
| | 46–60 | 21 (21%) | 26 (21.5%) | 47 (21.3%) |
| Education of the respondents | Illiterate | 13 (13%) | 13 (10.7%) | 26 (11.8%) |
| | Grade 1–8 | 33 (33%) | 57 (47.1%) | 90 (40.7%) |
| | Grade 9–12 | 34 (34%) | 40 (33.1%) | 74 (33.5%) |
| | Above grade 12 | 20 (20%) | 11 (9.1%) | 25 (11.3%) |
| Do you have experience in poultry production? | Yes | 65 (65%) | 113 (93.4%) | 178 (80.5%) |
| | No | 35 (35%) | 8 (6.6%) | 43 (19.5%) |

### 3.2. Characteristics of Chicken Farms

In this survey, three types of farms were visited depending on the type of chicken reared in the farms. These were farms rearing only egg-type (156 (70.6%)), broiler-type (30 (13.6%), and both egg-type and broiler-type chickens (35 (15.8%)). Regarding the chicken age groups, 18 (8.1%) farms keep breeders to produce day-old chickens, 66 (29.9%) sell pullets at the age of 3 months, 19 (8.6%) sell broilers at 45–60 days, 114 (51.6%) sell table eggs to market, and 4 (1.8%) of the farms rear both broiler and layer chickens and sale broilers and table eggs. Among the farm owners, only 43 (19.5%) know the weight of birds, and only 32.6% know the weight of eggs they are producing at the sale. The mean flock sizes, annual production capacity pullets, broilers, and layers of the farms were 925.3 ± 924.2, 2603.8 ± 2241, 2133.6 ± 1892.8, and 763.2 ± 809.8, respectively, with significant variability among the study areas are shown in Table 2. In most of the farms, 212 (95.5%) were reared in deep litter housing, and only 9 (4.1%) were in their own cage system. The bedding materials in the poultry houses were straw, sawdust, and wooden shaving, with a proportion of 170 (76.9%), 23 (10.4%), and 17 (7.4%), respectively. The majority of poultry buildings (36.2%) have corrugated iron sheet walls, followed by hardwood walls (34.4%) and brick walls (29.4%) in terms of structure. The chicken buildings' floors were made of concrete (51.1%), bare ground (45.7%), and laminated wood (3.2%). To control air

circulation in the poultry houses, most of the farm buildings 210 (95%) have openings for ventilation, and very few farms, 11 (5%), were without openings. The majority of the farms, 137 (62%), had their own working place, and the rest operated in a rental place. Among the farm owners, 171 (77.4%) exercised "All-In and All-Out" strategies for flock restocking and destocking.

**Table 2.** Production cycle of different classes of chickens, annual production, and farm flock size.

| Variables | Arsi Zone | | East Showa | | Total | |
|---|---|---|---|---|---|---|
| | Mean ± SD | Range | Mean ± SD | Range | Mean ± SD | Range |
| Production cycle of pullets/year | 3.0± 0.71 | 2–4 | 2.9 ± 0.97 | 1–6 | 2.98 ± 0.82 | 1–6 |
| Annual production of pullets/year | 1803.3 ± 1226.7 | 100–4800 | 3804.5 ± 2850.5 | 460–9000 | 2603.8 ± 2241 | 100–9000 |
| Production cycle of broiler/year | 2.1 ± 0.35 | 2–3 | 2.9 ± 0.90 | 1–5 | 2.76 ± 0.88 | 1–5 |
| Annual production of broilers/year | 1311.1 ± 534.9 | 300–2000 | 2314.2 ± 2036.2 | 150–6000 | 2133.6 ± 1892.8 | 150–6000 |
| Current flock size of all chicken | 741.1 ± 801.1 | 50–4000 | 1077.5 ± 992.4 | 50–4000 | 925.3 ± 924.2 | 50–4000 |
| Flock size of layers | 697.8 ± 638.8 | 50–2500 | 818.1 ± 930 | 50–3500 | 763.2 ± 809.8 | 50–3500 |

### 3.3. Poultry Disease Management Practices

The first, second, and third economically important poultry diseases in the farms were Newcastle disease (NCD), 140 (79.5%), infectious bursal disease (*Gumboro*), 58 (54.7%), and fowl cholera, 69 (88.5%), respectively as indicated by poultry producing farmers. However, there were also other important poultry diseases that were frequently observed in chicken farms, such as coccidiosis and fowl pox. In order to control disease outbreaks in poultry farms, 191 (86.4%) farm owners vaccinate their chickens for different diseases such as NCD, *Gumboro*, Marek's, and fowl pox diseases. However, only 157 (71.0%) of the farm owners follow standard vaccination schedules in terms of frequency. Among the farm owners, 65 (34%) and 58 (30.4%) vaccinate their chickens once and twice, respectively. Most of the farms, 137 (62%), provide treatment for sick birds, and in most cases, 142 (64.3%) of the farms faced disease outbreaks that caused serious mass death of birds. The mean annual occurrence of disease outbreaks in the study farms was 0.61 ± 1.0, with significant variability among the study areas. Depending on the farms' flock size, there is a continuous monthly and yearly bird death reports. The minimum and maximum annual death of chickens recorded was 1 and 150 chickens, with significant variability among the study areas Table 3.

**Table 3.** Monthly and yearly mortality of chickens in the farm and mass death of chickens due to outbreak.

| Variables | Arsi Zone | | East Show Zone | | Total | | *p*-Value |
|---|---|---|---|---|---|---|---|
| | Mean ± SD | Range | Mean ± SD | Range | Mean ± SD | Range | |
| Monthly mortality | 6.74 ± 5.17 [a] | 1–20 | 4.91 ± 4.48 [b] | 1–15 | 5.93 ± 4.94 | 1–20 | 0.025 |
| Annual mortality | 32.31 ± 28.39 [a] | 2–120 | 30.27 ± 33.75 [a] | 1–150 | 31.2 ± 31.3 | 1–150 | 0.644 |
| Mass death by outbreak at a time | 100 ± 80.5 | 50–300 | 79.38 ± 50.53 | 50–200 | 89.69 ± 67.32 | 50–300 | 0.293 |

Values with different supper-scripts (a,b) indicated statistica differences between the colomns & same letter indicated no differences.

### 3.4. Biosecurity Evaluation

Biosecurity refers to actions and measures implemented to prevent and control the introduction and spread of infectious diseases causing agents to a flock in a farm [7]. Biosecurity can be applied in three stages such as isolation which deals with protecting chickens from sources of infection, traffic control which involves limiting traffic movement, and

controlling sanitation which is about cleaning and limiting movements of equipment [13]. Others stated that biosecurity measures have conceptual, structural, and operational frameworks which involve housing design and construction with management procedures that keep the flock free from infectious diseases [7,10,12].

### 3.4.1. Conceptual Biosecurity

In order to assess conceptual biosecurity status, 13 indicators were included in the questionnaire, and their frequencies and percentages of responses are given in Table 4. Nearly 86.9% of the farms were located near residential areas at a distance <200 m, and 53.4% of the farms were close to main public roads at a distance <500 m, which predisposes the farms to frequent noise, environmental, physical, and chemical contamination that discomforts the chickens. A significant number of farm owners (68.8%) have no training on the biosecurity concept, and 62% of the farms do not have disease management record books.

**Table 4.** The frequency and percentage of indicators of conceptual biosecurity.

| Biosecurity Indicators | Category | Number of Farms | Percentage |
|---|---|---|---|
| | <50 m | 89 | 40.3 |
| Distance from residential Area (m) | (50–200) m | 103 | 46.6 |
| | >200 m | 29 | 13.1 |
| | <200 m | 46 | 20.8 |
| Distance from main road (m) | (200–500) m | 72 | 32.6 |
| | >500 m | 103 | 46.6 |
| Distance from nearest farm (m) | <500 m | 65 | 29.4 |
| | ≥500 m | 156 | 70.6 |
| Is there no standing water near your farm | Yes | 192 | 86.9 |
| | No | 29 | 13.1 |
| Poultry house with good ventilation | Yes | 164 | 74.2 |
| | No | 57 | 25.8 |
| Poultry house orientation | East–West | 143 | 64.7 |
| | North–South | 78 | 35.3 |
| Do the poultry house water/moisture proof? | Yes | 121 | 54.8 |
| | No | 100 | 45.2 |
| Biosecurity training to employee | Yes | 69 | 31.2 |
| | No | 152 | 68.8 |
| No more than one farm gate | Yes | 189 | 85.5 |
| | No | 32 | 14.5 |
| Maintaining records for diseases management | Yes | 84 | 38.0 |
| | No | 137 | 62.0 |
| No Management of sick animals after healthy ones | Yes | 32 | 14.5 |
| | No | 189 | 85.5 |
| Availability of visitors' logbook | Yes | 0 | 0.00 |
| | No | 121 | 100.0 |
| Having poultry production experiences | Yes | 157 | 71.0 |
| | No | 64 | 29.0 |

The conceptual biosecurity indicators revealed that the majority of the farms, 113 (51.1%), have a biosecurity score of <50%, which a is poor biosecurity level; 101 (45.7%) of the farms have a biosecurity score of 50–75%, and 7 (3.2%) of the farms have a biosecurity score of >75% demonstrating an aggregate of 48.95% of the farms have good BS. The mean conceptual BS for the farms was 50.4 ± 11.62, with minimum and maximum score values ranging from 23.08 to 84.62.

### 3.4.2. Structural Biosecurity

The structural biosecurity indicators are presented in Table 5. Most of the farms, 196 (88.7%), do not have tire baths for vehicle entry at the farm gate; 68.3% (151) of the farms do not control wild birds from accessing bedding materials, and 119 (53.8%) do not have access to information about disease outbreak at a regional and national level. In some farms, 102 (46.2%) do not have a standard quarantine house for newly incoming chickens. Pipe water (86.9%) and river water (13.1%) were the main sources of water for the farms. No farm has ever used water specifically microbiologically treated for chicken; instead, they have always used piped water that has been treated for human consumption. The mean structural BS of the farms was 63.27 ± 10.51, ranging from 35.29 to 82.35. The majority of the farms (70.6%) have a structural biosecurity score between 50 and 75%, 12.7% of the farms have structural biosecurity above 75%, and 12.2% have less than 50%. In general, in terms of structural biosecurity measurements, the majority of the farms, 194 (87.8%), are at good BS, having a score above 50%.

**Table 5.** The frequency and percentage of structural biosecurity indicators.

| Biosecurity Indicators | Category | Number of Farms | Percentage |
|---|---|---|---|
| Presence of fence and gate | Yes | 190 | 86.0 |
| | No | 31 | 14.0 |
| Presence of functional footbath | Yes | 144 | 65.2 |
| | No | 77 | 34.8 |
| Presence of only one vehicle entry point | Yes | 198 | 89.6 |
| | No | 23 | 10.4 |
| Presence of tire bath/spray at the gate | Yes | 25 | 11.3 |
| | No | 196 | 88.7 |
| Prohibition of entry of visitors | Yes | 173 | 78.3 |
| | No | 48 | 21.7 |
| No purchase of day-old chicken | Yes | 132 | 59.7 |
| | No | 89 | 40.3 |
| No purchase of feed | Yes | 0 | 0.00 |
| | No | 221 | 100.0 |
| No equipment exchanges with other farms | Yes | 204 | 92.3 |
| | No | 17 | 7.7 |
| No pet animal present in the farm | Yes | 136 | 61.5 |
| | No | 85 | 38.5 |
| Presence of permanent rodent control | Yes | 159 | 71.9 |
| | No | 62 | 28.1 |

**Table 5.** *Cont.*

| Biosecurity Indicators | Category | Number of Farms | Percentage |
|---|---|---|---|
| Presence of permanent wild bird control | Yes | 136 | 61.5 |
|  | No | 85 | 38.5 |
| No access to stored fresh litter for wild birds | Yes | 70 | 31.7 |
|  | No | 151 | 68.3 |
| No access to stored food for wild bird | Yes | 185 | 83.7 |
|  | No | 36 | 16.3 |
| No feeding of chicken outside | Yes | 209 | 94.6 |
|  | No | 12 | 5.4 |
| Well informed regarding poultry disease outbreak in the area | Yes | 102 | 46.2 |
|  | No | 119 | 53.8 |
| Surface water not used for drinking chicken | Yes | 192 | 86.9 |
|  | No | 29 | 13.1 |
| Do you have quarantine for new incoming flocks | Yes | 119 | 53.8 |
|  | No | 102 | 46.2 |

### 3.4.3. Operational Biosecurity

The operational biosecurity measurements are presented in Table 6. Most of such measurements were not implemented by the farms. Most of the employees do not wear special farm clothes such as shoes (50.7%), hand gloves (67.4%), mouth/nose masks (85.1%), and head hats (69.2%). The majority (90.9%) of the farms do not have visitors' cloth, and 71% do not have proper dead bird disposal places and procedures. The result revealed that 146 (66.1%) of farms have an operational BS less than 50%, 73 (33.0%) of the farms have scores in a range of 50–75%, and 2 (0.9%) of the farms have scores above 75%. The mean operational BS of the farms was 44.69 ± 13.04, ranging from 16 to 84.

**Table 6.** The frequency and percentage of operational biosecurity indicators.

| Biosecurity Indicators | Category | Number of Farms | Percentage |
|---|---|---|---|
| Employee use of special cloth (coveralls) | Yes | 145 | 65.6 |
|  | No | 76 | 34.4 |
| Employee use of special footwear (boots) | Yes | 109 | 49.3 |
|  | No | 112 | 50.7 |
| Employee use of hand glove | Yes | 72 | 32.6 |
|  | No | 149 | 67.4 |
| Employee use of special masker | Yes | 33 | 14.9 |
|  | No | 188 | 85.1 |
| Employee use of special hat | Yes | 68 | 30.8 |
|  | No | 153 | 69.2 |
| Culling of different class of chickens | Yes | 120 | 54.3 |
|  | No | 101 | 45.7 |
| Shower in and out | Yes | 40 | 18.1 |
|  | No | 181 | 81.9 |

**Table 6.** *Cont.*

| Biosecurity Indicators | Category | Number of Farms | Percentage |
|---|---|---|---|
| Visitors' use of special cloth | Yes | 20 | 9.1 |
| | No | 201 | 90.9 |
| Not keeping multiple ages together | Yes | 203 | 91.9 |
| | No | 18 | 8.1 |
| Partial depopulation | Yes | 50 | 22.6 |
| | No | 171 | 77.4 |
| Presence of paved places of discharge | Yes | 9 | 4.1 |
| | No | 212 | 95.9 |
| Regular cleaning and disinfection | Yes | 98 | 44.4 |
| | No | 123 | 55.7 |
| Used cleaning water is not drained outside | Yes | 82 | 37.1 |
| | No | 139 | 62.9 |
| High-pressure sprayer used for cleaning | Yes | 18 | 8.1 |
| | No | 203 | 91.9 |
| Proper disposal of dead chickens | Yes | 64 | 29.0 |
| | No | 157 | 71.0 |
| Removed litter stored at cover shade | Yes | 85 | 38.5 |
| | No | 136 | 61.5 |
| Applying insecticide on top of new litter | Yes | 25 | 11.3 |
| | No | 196 | 88.7 |
| No access to stored food for rodents | Yes | 62 | 28.1 |
| | No | 159 | 71.9 |
| Presence of isolation room for sick chicken | Yes | 118 | 53.4 |
| | No | 103 | 46.6 |
| Regular examination of sick birds | Yes | 164 | 74.2 |
| | No | 57 | 25.8 |
| Calling veterinarian when chickens get sick | Yes | 185 | 83.7 |
| | No | 36 | 16.3 |
| Vaccinating chickens/recommendations | Yes | 129 | 58.4 |
| | No | 92 | 41.6 |
| Use of antibiotics/recommended dosage | Yes | 112 | 50.7 |
| | No | 109 | 49.3 |
| Presence of record-keeping | Yes | 118 | 53.4 |
| | No | 103 | 46.6 |
| No contact between farm and other farms | Yes | 221 | 100.0 |
| | No | 0 | 0.00 |

3.4.4. Overall Biosecurity of the Farms

In general, most of the farms under investigation, 131 (59.3%), have an overall BS above 50%, and as a result, they are at good biosecurity management practices. On the other hand, 90 (40.7%) have an overall BS < 50%, which means they are at poor biosecurity

management, which needs more interventions. The mean overall BS of the farm was $51.78 \pm 7.48$, ranging from 32.73 to 72.73.

### 3.5. Association between Biosecurity Level and Farm Characteristics

From the lists of farm characteristics considered, poultry production experience ($\chi^2 = 10.90$; $p = 0.001$), biosecurity training ($\chi^2 = 17.353$; $p = 0.000$), presence of isolation room ($\chi^2 = 24.553$; $p = 0.001$), proper disposal of dead birds ($\chi^2 = 4.546$; $p = 0.033$), and owning disease record books ($\chi^2 = 20.89$; $p = 0.000$) have statistically significant association with biosecurity level of the farm (Table 7). In addition, the occupation of farm owners ($\chi^2 = 9.708$; adjusted $p$-value = 0.006), education level of owners ($\chi^2 = 10.143$; adjusted $p$-value = 0.006), farms' flock size ($\chi^2 = 30.361$; adjusted $p$-value = 0.008), and farm distance from the main road ($\chi^2 = 8.674$; adjusted $p$-value = 0.008) have statistically significant relationship with farms' biosecurity level. Farms having <250 chickens have been graded as "poor" and farms owning >1000 chickens have been graded as "good" biosecurity level (Table 7). Farm owners' gender, the distance between farms, farm ownership (private, MSE, Cooperative), and type of chickens were not significantly associated with biosecurity level ($p > 0.05$).

**Table 7.** Association between biosecurity level and some farm characteristics.

| Variables | Categories | Biosecurity Status (%) | | Chi-Square Value | $p$ Value/Adjusted $p$-Value |
|---|---|---|---|---|---|
| | | Good (>50%) | Poor (<50%) | | |
| Owners' gender | Male | 96 (62.3%) | 58 (37.7%) | 1.973 | 0.160 [ns] |
| | Female | 35 (52.2%) | 32 (47.8%) | | |
| Poultry production Experience | Yes | 104 (66.2%) | 53 (33.8%) | 10.90 | 0.001 * |
| | No | 27 (42.2%) | 37 (57.8%) | | |
| Distance from another farm | <500 m | 35 (53.8%) | 30 (46.2%) | 1.125 | 0.289 |
| | >500 m | 96 (61.5%) | 60 (38.5%) | | |
| Biosecurity training | Yes | 55 (79.7%) | 14 (20.3%) | 17.353 | 0.000 * |
| | No | 76 (50.0%) | 76 (50.0%) | | |
| Farm premises | Own | 77 (56.2%) | 60 (43.8%) | 1.409 | 0.235 |
| | Rented | 54 (64.3%) | 30 (35.7%) | | |
| Having isolation room | Yes | 88 (74.6%) | 30 (25.4%) | 24.553 | 0.000 * |
| | No | 43 (41.7%) | 60 (583%) | | |
| Proper disposal of dead birds | Yes | 45 (70.3%) | 19 (29.7%) | 4.546 | 0.033 * |
| | No | 86 (54.8%) | 71 (45.2%) | | |
| Having a disease record book | Yes | 66 (78.6%) | 18 (21.4%) | 20.89 | 0.000 * |
| | No | 65 (47.4%) | 72 (52.6%) | | |
| Farm ownership | Private | 65 (53.3%) | 57 (46.7%) | 5.660 | 0.056 [ns] |
| | MSE | 57 (69.5) | 25 (30.5%) | | |
| | Cooperative | 9 (52.9%) | 8 (47.1%) | | |
| Major livelihoods of owners | Farmers | 75 (64.7%) | 41 (35.3%) | 9.708 | 0.006 * (0.087) |
| | Nonemployee | 13 (36.1%)[a] | 23 (63.9%) [b] | | (0.002) |
| | Employee | 26 (63.4%) | 15 (36.6%) | | (0.549) |
| | Trader | 17 (60.7%) | 11 (39.3%) | | (0.865) |

**Table 7.** *Cont.*

| Variables | Categories | Biosecurity Status (%) | | Chi-Square Value | p Value/Adjusted p-Value |
| --- | --- | --- | --- | --- | --- |
| | | Good (>50%) | Poor (<50%) | | |
| Education level of owners | Illiterate | 12 (46.2%) | 14 (53.8%) | 10.143 | 0.006 * (0.162) |
| | Grade 1–8 | 46 (51.1%) | 44 (48.9%) | | (0.046) |
| | Grade 9–12 | 54 (73.0%) [a] | 20 (27.0%) [b] | | (0.004) |
| | Above grade | 19 (61.3%) | 12 (38.7%) | | (0.841) |
| Farm flock size | <250 | 23 (34.8%) [a] | 43 (65.2%) [b] | 30.361 | 0.008 * (0.000) |
| | 250–1000 | 49 (59.8%) | 33 (40.2%) | | (0.920) |
| | >1000 | 59 (80.8%) [a] | 14 (19.2%) [b] | | (0.000) |
| Farm distance from main road | <1000 m | 36 (78.3%) [a] | 10 (21.7%) [b] | 8.674 | 0.008 * (0.004) |
| | 1000–2000 m | 39 (54.2%) | 33 (45.8%) | | (0.271) |
| | >2000 m | 56 (54.4%) | 47 (45.6%) | | (0.162) |
| Chicken type | Egg type | 91 (58.3%) | 65 (41.7%) | 0.802 | 0.670 [ns] |
| | Broiler type | 20 (66.7%) | 10 (33.3%) | | |
| | Meat and Egg | 20 (57.1%) | 15 (42.9%) | | |

Values with different supper-scripts (a,b) indicated statistical differences between the columns & same letter indicated no differences, * indicated significant statistical differences between two groups and where ns indicated non significances.

## 4. Discussion

The current study assessed the biosecurity status and its association with farm characteristics on private, MSE, and cooperative-based commercial poultry farms in the Arsi and East Showa zones of Ethiopia. Male-owned farms were dominant in the area, indicating gender inequalities in terms of access to finance, entrepreneurial capabilities, and socio-cultural impacts. Likewise, a study in Nigeria indicated that most of the farms were owned by males (86.4%), which is attributed to rigor, stress, and challenges that describe poultry production enterprises which many females might not be able to cope with [14]. A study conducted in the Bishoftu area of Ethiopia indicated 63.4% of commercial poultry farm owners were male [6]. Women have less access to education and are more affected by cultural issues which have a positive association with entrepreneurship. Most of the farm owners were found to have secondary education, where most of them (40.7%) were grade 1–8 and (33.5%) were grade 9–12. Such educational levels were similar to what was reported in Cameroon [15], reporting secondary education of farm owners. Though there is heterogeneity in the educational level of farm owners, education could impact the management and resource use efficiency of the farms. Previous studies in tropical areas indicated that the farm operator's educational level enhanced their ability to make use of information about production and market input and overall production efficiency [14,16].

The dominant poultry farms in the study area were the production of egg layers (70.6%), which includes layers, pullet, and breeder layer production business, followed by broiler (13.6%) and dual-type chicken (13.6%) production. A similar finding was reported in central Ethiopia, that 63.4% of the farms were layer-producing farms [6]. Most of the farms (62%) run their farm business on their own farmland, having different sizes depending on their location. The aggregate mean annual flock size of the farms, regardless of the farm type, was 916.2 ± 914.1, with a mean number of production cycles per year of 2.87 ± 0.85. The study conducted around the Debre Markos area of Ethiopia stated a farm flock size of 844.3 [17]. Similar flock sizes and production cycles were reported in Cameroon, with a size of 1181.37 ± 989.52 and 3.89 ± 1.10 [15].

Diseases were the most challenging constraint facing poultry production in Ethiopia, though there are other constraints. Poultry diseases are considered to be the most impor-

tant problem contributing to reducing both the number and productivity of chickens in Ethiopia [18]. The top three economically important poultry diseases in the commercial farms were Newcastle disease (NCD), 140 (79.5%), infectious bursal disease (Gumboro), 58 (54.7%), and fowl cholera, 69 (88.5%), respectively. A similar result was reported in Ethiopia [18,19]. Poultry disease was reported to be a critical challenge for poultry-producing farms in Cameroon [20]. The majority (86.4%) of commercial farms in selected districts of Arsi and East Showa zones practiced chicken vaccination; however, most of the vaccination schedule is below standard as 34% vaccinate once and 30% vaccinate twice for those diseases that require repeated vaccination. Proper vaccine utilization, access, poor biosecurity, and generally a lack of an organized poultry health service delivery system were the major bottlenecks for the poultry industry in the country [21].

The dominant poultry house was the deep litter type (95.5%), followed by the cage system (4.1%). Studies in Cameroon and Nigeria indicated most of the commercial poultry farms use deep litter poultry housing, 77.8% and 83.3%, respectively [20,22]. There were no pond or reservoir water sources (86.9%) near the farms as it is a source for disease outbreaks. In agreement with [6], who found 70.45% of farms were far from standing water sources in Ethiopia. A similar report in the UK indicated that 71.6% of backyard poultry producers revealed their chickens do not access pond water [23]. The level of awareness of biosecurity in small-scale commercial poultry farms in the Arsi and East Showa zones was very low, as 40.7% of farms have an overall BS < 50%. Inadequate awareness of biosecurity obstructed the proper implementation of biosecurity practices [24,25]. The conceptual, structural, and operational biosecurity frameworks proposed by [10,12] were taken into consideration in this study. Structural biosecurity practices were the most frequently implemented practices, as 87.8% of farms had BS over 50% in terms of structural biosecurity, while in terms of conceptual and operational biosecurity, only 48.95% and 33.9% of the farms had BS above 50%, respectively. The study revealed that most farms have fences (86%), footbaths at farm gates (65.2%), and prohibition of entry of visitors to the farm (78.3%), which were promising practices. This was lower than the finding of [6], who reported 90.91% of the farms to have footbaths, but it is in line with visitors' prohibition (70.45%) reported in the Bishoftu area. Similarly, this was lower than what was reported in the Mekelle area (80%) but similar to the prohibition of visitors' entry (76%) except for authorized visitors [26]. Regarding cleaning and disinfection, only 44% of the farms use disinfectant, which is by far less than what was reported in the Mekele area, which was 88% [26]. Such differences might be due to farm size and location of the farms, where the current study considered farms at the woreda level, where there is limited access to information, awareness, and inputs.

Most farms (77.4%) practiced all-in and all-out flock movement, which was higher than the 54.55% report in the Bishoftu area [6], which is encouraging as partial flock movement predisposes the farm to infectious diseases. Different studies pointed out that buying animals from different farms entails a greater risk of introduction of disease-causing agents [27]. Only 53.8% of the farm had isolation room to quarantine sick or newly coming chickens which is a risk to the full operation. A study conducted in Turkey disclosed that only 36% of producers keep disease records, and 44% of respondents used quarantine for new animals upon arrival [28]. Typical employee cloth was used by 65.6% of the farms in the current study, which was in line with the study conducted in Bishoftu town, which disclosed 65.91% [6] and 63.3% in the Debre Markos area [17]. The use of special employees' and visitors' cloth reduces the incidence of entry of diseases causing microorganisms into the farms from sources such as distant areas, household poultry, and from other farms. When uniforms and shoes were not provided to farm employees, the chance of wearing in-house clothes and shoes increased [22]. These clothes might have contact with local poultry at home or out of a home that picks an infectious agent and brings it to the farm [29]. In general, higher levels of biosecurity are associated with less prevalence and outbreak of poultry diseases.

Regarding the association between farms' BS and characteristics, in this study, gender, farm ownership type, farm premises, and farm type were not found to affect the level of

biosecurity adoption. Farm owners' poultry production experience, biosecurity training, presence of isolation room, proper disposal of dead birds, disease record, occupation, education level, farm flock size, and farm distances from the main road were found to affect BS in commercial farms. Studies indicated that farmers with larger flock sizes tended to have enhanced biosecurity practices [15,30]. Poor biosecurity score was recorded in farms owning smaller flock size, <250 chickens, compared to farms having more than 1000 chickens which might be related to higher negligence and lower commitment by owners in implementing biosecurity practices in the case of farms with fewer chickens. In line with the current study, the level of awareness of biosecurity practices was high in farms having larger flock sizes which improved the biosecurity score of the farms in Nigeria [10].

In the current study, farms located a distance above 2 km from the main road have poor biosecurity scores, unlike [9], who claimed farms located far from the main road have a better biosecurity level. This could be due to limited access to extension, veterinary services, training, information, and input supply as the farm gets far away from roads in countries such as Ethiopia, where infrastructure and facilities are less developed. The education of the farm owners impacted BS, where well-educated farmers improved their farms' biosecurity practices through better adoption rates. Education enhances the ability of farmers to analyze and understand biosecurity measures [30]. Farmers' primary occupation was significantly associated with BS, where farms owned by nonemployees have poor biosecurity scores, which might be due to financial limitations that reduced their commitment and focus. The occurrence of disease outbreaks in poultry farms decreased with increasing biosecurity scores, thereby supporting the relevance of biosecurity adoption to control diseases [10]. There have been reports of a reduction in infectious disease outbreaks with standard biosecurity protocols [12].

## 5. Conclusions

The practices of biosecurity are a fundamental footstep for preventing the introduction and spread of pathogenic microorganisms that initiate diseases in poultry farms. The study confirmed that most commercial poultry farms were handled by males at a small-scale level practicing under low biosecurity scores below average. The ineffective application of biosecurity procedures revealed the need for a comprehensive capacity-building program, information dissemination, and awareness raising among farming communities because the consequences of the biosecurity issue are extremely severe and result in total losses through the outbreak of diseases.

Additionally, the lack of biosecurity on commercial farms results in a higher prevalence of diseases, extensive drug use, high levels of drug resistance, increased costs, chicken deaths, and, ultimately, drug residues in chicken products that may be important for public health. Most of the farm characteristics have a positive association with biosecurity measurements which indicates that the poor biosecurity score of most of the farms could be improved by improving the production system that, ultimately, boosts the farms' productivity and reducing the economic impacts of poultry diseases. Much effort, energy, and costs need to be spent in the area of farm site determination, traffic control, and operational biosecurity measurements implementation and practices to improve the scenario.

**Author Contributions:** Conceptualization, D.T., B.T., and G.G.; Data curation, D.T.; Formal analysis, D.T.; Investigation, D.T.; Methodology, B.T. and G.G.; Resources, B.T. and G.G.; Software, D.T.; Supervision, D.T., B.T., and G.G.; Validation, D.T., B.T., and G.G.; Visualization, B.T. and G.G.; Writing—original draft, D.T.; Writing—review and editing, B.T. and G.G. All authors have read and agreed to the published version of the manuscript.

**Funding:** In this study: we have not received any specific funds. Instead, Addis Ababa University has supported the corresponding author as a Ph.D. student as per the national postgraduate support guidelines.

**Institutional Review Board Statement:** The current study does not involve either human beings or animals directly for data and sample collection. Rather, the study used a cross-sectional survey type that addressed through interviews and observations after being obtained informed consent from the respondents and farm owners.

**Informed Consent Statement:** The study was reviewed at the department and college level by professionals from animal and veterinary sciences, including the research approval committee of Arsi University. The study does not involve animals but involved farm owners on a voluntary basis having their informant consent after briefing the objectives of the study and the data to be recollected. The respondents were told that all information about farms is kept secret, no names or any personal identifier information will be mentioned in public, and the information is only used for research and publication purposes.

**Data Availability Statement:** For this research, we have tried to collect primary data from commercial poultry farms in two zones, and hence, it is available at the hand of the researchers. The primary data can be available at any time.

**Acknowledgments:** This work was supported by Addis Ababa University, Ethiopia. We thank all commercial poultry farms in the study areas for their provision of all essential data for the success of this research.

**Conflicts of Interest:** The authors declare no competing interests.

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
