# Peer review of "Assessment of Biosecurity Practices and Its Status in Small- and Medium-Scale Commercial Poultry Farms in Arsi and East Showa Zones, Oromia, Ethiopia"

_poultry, doi:10.3390/poultry2020025_

Round 1
Reviewer 1 Report
This is a very interesting and timely study. It highlights the importance of biosecurity and brings attention to the lack of Extension workers capable of providing training on biosecurity and other areas. Focuses attention on the fact that much more needs to be done in the area of trainings related to disease prevention and biosecurity.
This is a very interesting article on biosecurity with some valuable information that highlights the importance of biosecurity training.
Line 13 – add “s” to outbreak
Line 17 – replace “like” with “including”
Line 24 – replace “shown” with “indicated”
Line 33 – insert “is” between “and” and “highly”
Line 35 – replace “as” with “at”
Line 39 – delete “one”
Line 41 – remove the extra space between “was” and “about”
Line 45 – insert “a” between “plays’ and “crucial”
Line 50 – insert “for” between “accounts” and “about”
Line 51 – remove the extra space between “as” and “draught”
Line 56 – replace “were” with “are”
Line 58 – remove extra space between “respectively” and “which”
Line 59 – replace “chickens death” with “mortality rate”
Line 60 – remove the extra space between “egg” and “and”
Line 64 – replace “constraints” with constraint”
Line 65 – replace “like” with “such as”
Line 66 – replace “cause” with “causes”
Line 67 – remove the extra space between “in” and “all”
Line 95 – replace “requires” with “require”
Line 107 – replace “are” with “were”
Line 113 – delete “a”
Line 131 – replace “know” with “determine”
Line 142 – replace “ with .
Line 146 – replace “got” with “gotten”
Line 148 – replace “obtained” with “obtain”
Line 156 – replace “sale’ with “sell”
Line 157 – replace “sale” with “sell”
Line 158 – replace “sale” with “sell”
Line 165 – replace ‘shivering” with “shaving”
Line 176 – insert “were” between “which” and “frequently”
Line 183 – replace “farm” with “farms”
Line 194 – remove extra space between “as” and “isolation”
Line 196 – replace “controlling which is about limiting” with “controlling sanitation which is about cleaning and limiting”
Line 270 – replace “has” with “have”; insert “are” between “and” and “more”
Line 276 – replace “area” with “areas”
Line 287 – remove the extra space between “size” and “and”
Line 291 – replace “for” with “to”
Line 294 – replace “report” with “reported”
Line 295 – remove the extra space between “be” and “a”
Line 305 – insert “is” between “it” and “a”
Line 330 – insert “to the” between “risk” and “full”
Line 333 – insert “with the” between “inline” and “study”
Line 338 – replace “closes” with “clothes”
This is a very interesting and timely study. It highlights the importance of biosecurity and brings attention to the lack of Extension workers capable of providing training on biosecurity and other areas. Focuses attention on the fact that much more needs to be done in the area of trainings related to disease prevention and biosecurity.
This is a very interesting article on biosecurity with some valuable information that highlights the importance of biosecurity training.
Line 13 – add “s” to outbreak
Line 17 – replace “like” with “including”
Line 24 – replace “shown” with “indicated”
Line 33 – insert “is” between “and” and “highly”
Line 35 – replace “as” with “at”
Line 39 – delete “one”
Line 41 – remove the extra space between “was” and “about”
Line 45 – insert “a” between “plays’ and “crucial”
Line 50 – insert “for” between “accounts” and “about”
Line 51 – remove the extra space between “as” and “draught”
Line 56 – replace “were” with “are”
Line 58 – remove extra space between “respectively” and “which”
Line 59 – replace “chickens death” with “mortality rate”
Line 60 – remove the extra space between “egg” and “and”
Line 64 – replace “constraints” with constraint”
Line 65 – replace “like” with “such as”
Line 66 – replace “cause” with “causes”
Line 67 – remove the extra space between “in” and “all”
Line 95 – replace “requires” with “require”
Line 107 – replace “are” with “were”
Line 113 – delete “a”
Line 131 – replace “know” with “determine”
Line 142 – replace “ with .
Line 146 – replace “got” with “gotten”
Line 148 – replace “obtained” with “obtain”
Line 156 – replace “sale’ with “sell”
Line 157 – replace “sale” with “sell”
Line 158 – replace “sale” with “sell”
Line 165 – replace ‘shivering” with “shaving”
Line 176 – insert “were” between “which” and “frequently”
Line 183 – replace “farm” with “farms”
Line 194 – remove extra space between “as” and “isolation”
Line 196 – replace “controlling which is about limiting” with “controlling sanitation which is about cleaning and limiting”
Line 270 – replace “has” with “have”; insert “are” between “and” and “more”
Line 276 – replace “area” with “areas”
Line 287 – remove the extra space between “size” and “and”
Line 291 – replace “for” with “to”
Line 294 – replace “report” with “reported”
Line 295 – remove the extra space between “be” and “a”
Line 305 – insert “is” between “it” and “a”
Line 330 – insert “to the” between “risk” and “full”
Line 333 – insert “with the” between “inline” and “study”
Line 338 – replace “closes” with “clothes”
Reviewer 2 Report
The authors should better describe the structures where the birds are reared, if they are prefabricated or fixied structures, with bricks or wood.
Moreover if the water has a microbiological control
Reviewer 3 Report
The manuscript "Assessment of Biosecurity Practices and Its Status in Small and Medium Scale Commercial Poultry Farms in Arsi and East Showa Zones, Oromia, Ethiopia" focus an important subject related to poultry production and their impact in the “One Health” perspective. The study assessed the biosecurity status and its association with poultry farm characteristics in Ethiopia.
The Abstract is correct and elucidates the content of the manuscript.
The introduction section is satisfactory. It gives a good framing of poultry production in Ethiopia, even the Biosecurity concepts and practices are not very developed.
The materials and methods seem adequate. In my opinion, the categories of questions (conceptual, structural, and operational biosecurity factors) and the questionnaire should be cleared in this section. The study implies vast statistical work with different variables, that must be considered. Table 1 is unnecessary.
The results section should be improved.
In Table 2, “Marital status of the respondent” is unnecessary. The Title of Table 2 should be correct, since there is no information regarding Religious characteristics. From Table 3, only data from farm flock size is used in the study and compared with other variables. I wonder the usefulness of table 3.
The discussion seems correct.
The conclusions are consistent with the evidence and arguments presented.
The English language should be revised.
“Error! Reference source not found.” appears several times along the text.
Round 2
Reviewer 2 Report
corrections to the work are accepted and thanks to the authors
Reviewer 3 Report
As reviewer, I congratulate that most suggestions were accepted and included in this new version of the manuscript "Assessment of Biosecurity Practices and Its Status in Small and Medium Scale Commercial Poultry Farms in Arsi and East Showa Zones, Oromia, Ethiopia", which improves substantially the manuscript.
The correction of English spelling enhanced the perception and interpretation of the content.
Nevertheless, I have some more “minor” suggestions.
The Abstract is correct and elucidates the content of the manuscript.
The introduction section improved substantially with the inclusion of Biosecurity concepts and practices.
The materials and methods seem adequate.
Lines 125-126: Remove the sentence “. In summary, comprehensive closed-ended questions were developed for this study.”, since this information is presented in lines 128-130.
Line 143: Remove “Then” at the beginning of the sentence “Then the computed biosecurity scores (BS) were compared with the standard biosecurity rating “Good” if BS of the farm was above 50% and “Poor” if the BS of the farm was less than 50%”.
The results section should be improved.
Lines 157-158: Remove this part of the sentence “…which impacted the success of a business and entrepreneurial skill of individuals. Since this is an explanation that should be in discussion. Remove ” at the end of the sentence.
Lines 160-163: Remove this sentence “Lifetime business experience determines the success, competition, and sustainability of any business through enhancing decision-making and management quality.” since is an interpretation that should be in the discussion section. Is not a result of your study.
Line 167: “3.2. Characteristics of chicken farms”. Correct letter size and/or style.
Line 170: Correct “…[35(15.8%).” to …[35(15.8%)].
Line 171: Correct “age groups 18(8.1%)” to …age groups, 18(8.1%).
Line 178: Remove ““Error! Reference source not found.”” I think that this appears every time the authors insert the reference to Tables in the text. As it appears several times in the results section. Lines 207, 221-222, 237-238, 273, 279.
Line 179: Correct “… and only 9(4.1%) of own cage system.” to … and only 9(4.1%) in own cage system.
Line 185: Correct “… most of the farm owners 210(95%) have openings…” to …most of the farm buildings 210(95%) have openings…
Lines 210-218: Those sentences should be implanted in Introduction section (near lines 71-77) because those are not descriptions of materials and methods.
Tables 4 and 5: why the variable “No more than one farm gate” is in table 4 and “Presence of fence and gate” is in table 5. They are not related variables that should be in the same table?
Line 241: Correct “In some farms, …” to Some farms, …
Line 268: “3.5. Association between biosecurity level and farm characteristics” Correct letter size and/or style.
Line 275: Correct “…farms’ flock size owners…” to …farms’ flock size…
The discussion seems correct.
Lines 335-336: “Structural biosecurity practices were the most frequently implemented practices than others, as…” Correct English spelling.
The conclusions are consistent with the evidence and arguments presented.
Minor editing of English language required.
See comments and suggestions.
